# Youth Practices of Reading as a Form of Life and the Digital World

Anna Shutaleva [1,2,*] , Ekaterina Kuzminykh [3] and Anastasia Novgorodtseva [3]

1 Department of Philosophy, Ural Federal University Named after the First President of Russia B. N. Yeltsin, 620002 Ekaterinburg, Russia

2 Department of Socio-Humanitarian Disciplines, Ural State Law University Named after V. F. Yakovlev, 620137 Ekaterinburg, Russia

3 Department of Applied Sociology, Ural Federal University Named after the First President of Russia B. N. Yeltsin, 620002 Ekaterinburg, Russia

* Correspondence: ashutaleva@yandex.ru; Tel.: +7-932-60331-73

**Abstract:** The proliferation of digital technologies is precipitating a transformation in the socio-cultural fabric of human existence. The present study is dedicated to investigating the coexistence of various reading practices among contemporary youth in the modern era. The advent of new forms of reading has resulted in a shift from conventional paper-based reading to electronic formats, which, in turn, has transformed the practice of reading and the way of life associated with it. The methodological foundation of this research is the socio-philosophical theory that the practice of reading, rooted in the habitus of reading, is enacted by practitioners, and organized through public initiatives. The context of the reading practice system is a distinct historical system of circumstances in which practices are reproduced. This study encompasses an empirical component, focusing on the examination of reading practices among young individuals in a large modern city, specifically within the confines of Yekaterinburg ($N = 200$). The research was conducted between December 2021 and January 2022. This study permits an analysis of the constituent elements of the reader's habitus model as a form of life.

**Keywords:** text; printed book; e-book; reading; habitus; youth; digital technologies; digital world





## 1. Introduction

Reading has always been considered a way for people to acquire culture [1–3]; however, how people read and consume information has evolved with the rise of digital media [4–6]. This statement emphasizes that the language is saturated with cultural themes and vice versa. The cultural practice of the community does not separate from its linguistic practice. People follow certain forms of life, the foundations of which organize the world and language, their cultural practices, and values. This idea is based on the belief that human behavior is not arbitrary but rather follows patterns that are shaped by cultural norms and traditions [7–9]. The foundations of these cultural patterns are rooted in language. Language is the primary means by which cultural practices are communicated and passed down from generation to generation. It is through language that people learn the customs and values of their society, and it is through language that they express their own beliefs and experiences [10–12]. Language shapes how people think about the world, and it influences their behavior subtly and profoundly.

Cultural practices are another important foundation of human behavior. These practices include everything from religious rituals to social customs to artistic expressions. They provide a framework for social interaction and help to define a society's identity [5,10,11]. Cultural practices are often deeply ingrained in people's lives, and they play a critical role in shaping individual and collective behavior. The foundations of cultural patterns are not fixed or unchanging. They constantly evolve in response to social, political, and

economic changes. Advances in technology have led to new forms of communication, social interaction, and the practice of reading, which are shaping the way people think about the world and their place in it.

Studying cultural patterns is critical because it provides insights into how people make sense of the world and their place in it. People follow certain forms of life that the foundations of language, cultural practices, and values shape. These foundations provide a framework for human behavior and shape the way people think about the world. The analysis of cultural patterns and how reading practices are being transformed as a cultural pattern is an essential area of research that can provide insights into the factors that shape human behavior and how cultural norms, and traditions evolve.

This article is devoted to the study of the coexistence of various reading practices of young people in the modern world. The modern world and the development of digital technologies provide people with access to various sources of information. Youth is the most mobile part of society, responding to innovations [13–15]. This circumstance influenced the choice of the object of research and focused it on the study of factors influencing the reading practices of young people.

People are initially placed in a semantic field, in a language game, following L. Wittgenstein can be defined as a single whole of language and actions with which language is intertwined [16]. "To speak the language" means to accept the form of life, the ways of manifestation and speaking, and thus the ontological foundations of human life in society. The form of life manifests itself at the level of semantics [17–19] because the boundaries of language are the boundaries of the human world and society.

The consumption of information in the information society is becoming a daily habit [20–22]. People can read messages, blogs, e-books, and articles just by going online, but this situation raises questions about what exactly counts as reading. When studying reading as a form of human life, the question "What do they read? Whom are they reading?" Now a third question can be added to these questions: "How do they read?". The study of reading practices is especially relevant at present when the place of a book in the world of digital technologies is rather ambiguous. At the same time, one of the significant pedagogical issues refers to philosophy in its essence: how to teach to read social and cultural meanings in the text? This question refers to one of the current trends in modern education, namely, the development of systemic and critical thinking [23–25], which is especially significant in the world of media communications.

Interest in the practice of reading is because it manifests self-awareness and ways of self-organization of people in society [26–28]. Reading practices are expressed in their synchronous-diachronic context, where stable components of society's values are reproduced in the lively activity of their questioning and interpretation. The process of reading is existential. The creative aspect of reading is recognized by J.-P. Sartre presented reading as a synthesis of perception and creativity since reading simultaneously assumes the essentiality of both the subject and the object [29]. R. Barthes defines reading as a process that complements the process of writing, which has the benefits of the original creative act [30]. U. Eco writes about the reader as a type of text strategy with its creative and active beginning of the book [31]. Therefore, the affirmation of the cultural communicativeness of reading is possible in the social-behavioral aspect and the ontological aspect.

The modern world is a world of digital technologies, which are changing the way society, culture, and communication exist [32–34]. Anonymity is becoming a condition of security in digital society [35–37]. The problem is that in the digital world, people can become another subject of knowledge along with other subjects. However, human consciousness is an existential position from which everything else is perceived and comprehended objectively. Also, the human environment and the objects of culture created by it cannot live according to those laws that are defined as natural. The reason is that natural objects last without human intervention. This provision does not apply to cultural phenomena and hence to the practices of reading a text.

Culture is an exceptional condition for the existence of people and their generations. Culture lasts and will be renewed only by the constant effort of people. The existence of cultural traditions is possible subject to the existence of people for whom they are an integral element of existence [38–40]. Cultural tradition involves the reproduction of basic cultural themes that bind humanity, which have extra-historical super value; that is, patterns. R. Benedict turned to the original biological term "pattern" in "Models of Culture" [41]. R. Benedict introduces the term "cultural patterns", which presents the dominant internal principles that ensure the commonality of cultural behavior in various spheres of human life. "Cultural patterns" allow representing people through their life in culture, and culture—through its manifestation in people, through the cultural conditioning of human reaction. Even spontaneous behavior can be thought of as a culturally determined response that makes up a large part of the vast stock of human behavior patterns.

The post-structuralist concept of communication as a sign exchange allows the interpretation of a sign as a "place" that links other signs and allows the exchange of signs and indications in the place of its presence [42–44]. As a result, the text turns out to be, in fact, the communication itself, the exchange or the place that creates the possibility of exchange and produces an exchange. The stability of reproducible cultural models in communicative practice depends on the maintenance of stable, rhythmically repeated sign-symbolic acts of communication through the signification of ongoing everyday life. Even communication becomes a space of reading, in which the symbolic space is constituted in the intersection and combination of endless interpretations of texts [45–47]. The texts are included in the matrix of understanding, which implies the unity of the semantic field and the culturally determined practice of reading texts, which is significant for modern anthropological studies [48–50].

The development of digital technologies is a kind of "challenge" for a book in a traditional format. Young people's reading practices have significantly changed in the modern digital world. While digital media has led to new content consumption, reading remains an essential way for young people to acquire culture and develop critical thinking skills [51–53]. However, there are concerns about the impact of digital media on young people's reading practices, and educators and parents must work to promote healthy reading habits and provide opportunities for young people to engage with high-quality content. However, there are concerns about the impact of digital media on young people's reading practices. Some experts worry that the constant stream of information and distractions from digital media may make it harder for young people to focus on reading and develop deep reading skills [54–56]. Others worry that the rise of digital media may lead to a decline in the quality and depth of the content that young people are reading [57–59].

The habit of modern people reading from the screen changes our way of thinking [60–62]. One of the most significant changes in people's reading practices is the shift toward digital reading. With the widespread availability of smartphones, tablets, and e-readers, young people can now access books, articles, and other forms of written content on various digital devices. This circumstance has led to a decline in traditional print media, such as newspapers and books, and an increase in digital content consumption. Another important change in young people's reading practices is the rise of social media and online platforms. Social media platforms have become important sources of information and news for young people. Many young people use these platforms to follow news outlets and other sources of information and to share and discuss content with their peers.

Reading from a phone screen is convenient, but it generates several challenges that can negatively affect the personality [63–65]. At the same time, there is an alternative point of view that e-books will never replace printed books [66]. The Internet is different from television, which was once also associated with the reading crisis, as the Internet is the technology that supports books [67–69]. The development of the Internet leads to the emergence of a new space for reading and new practices [70–72], which complement traditional practices and form a fluid, complex, and evolving system of reading.

The pragmatic turn in the social sciences led to new directions in research, first in philosophy and then in sociology, for example, in the works of P. Bourdieu [72], as well as the research of C. Geertz [73]. A distinctive feature of this approach is the consideration of routine, habitual actions, an appeal to the sociology of everyday life, as well as its compromise between objective social conditions and subjectivism, emphasizing the active role of the individual in the reproduction and change of the social system.

According to the theory of P. Bourdieu, practice is understood as everything that a social agent does himself and what he encounters in the social world [72]. Practice is reproduced by the agent; that is, it depends on its characteristics, it is reproduced within the framework of objective and subjective structures, and that is, it is also determined by external conditions. Thus, the study of reading practices needs to consider the peculiarities of the reader and the peculiarities of the social reality in which the studied practices unfold.

Background and context are important categories in the analysis of external conditions. The context includes environmental factors that are specific to a given place and time and are important for understanding the phenomenon under study. The concept of the background is borrowed from Gestalt psychology and comes from the idea of a figure and the background itself, which functions as a condition that gives meaning to the figure. The background is a set of practices with a spatiotemporal organization in which the individual is socialized.

P. Bourdieu defines habitus as systems of stable and transferable dispositions, structured structures predisposed to function as structuring structures [72]. That is, habitus is principles that generate and organize practices and ideas that, although they can be objectively adapted to their goal, do not imply a conscious focus on it and the indispensable mastery of the necessary operations to achieve it. Habitus is embodied in sets of practices familiar to the individual, typical patterns of behavior in each situation, the use of which is not always conscious. Habitus is formed under the influence of a certain context and is set by the structure of society by objective social conditions, the study of which can reveal both stable and unstable structures of the social order [74–76]. At the same time, habitus is characterized by the active ability of the individual to make changes to existing units. Thus, the reader's habitus is determined by the society in which the reader is located, while at the same time, individuals have a certain freedom in choosing reading practices.

According to P. Bourdieu, habitus can be not only an individual but also a group [72]. At the same time, of course, it is impossible to create the same conditions for all representatives of the group, but in this case, the similarity of social conditions of formation is allowed, which unites individual habitus of various members of one class with the proviso that they are structural possibilities of others and express the singularity of the position within the class, group. In this study, we consider young people as agents of practices whose reading practices are influenced by objective and subjective factors of reading habitus [77–79]. Modern organizations of reading practices are implemented both in traditional institutions (book publishers, bookstores, libraries in their traditional form) and in new ways (digital gadgets, the Internet, digital libraries, and stores, reading applications).

## 2. Materials and Methods

This article is devoted to the study of the coexistence of various reading practices. Of particular interest is the role that books and reading play in the lives of today's youth, growing up and maturing in the new information society. To analyze the reading practices of young people, a reader habitus model was developed, and an empirical study was carried out. To achieve the set goals, a methodological approach to the study of social practices proposed by P. Bourdieu [72], the concept of reading habitus [72,80,81], the sociocultural influencing of reading [82–84], the impact of digital technologies on the practice of education [85–87] was chosen.

This study reveals the following indicators of reader habitus:

— Objective indicators of reader habitus that are inherent in society and exist in the background category: reading books in childhood (with parents or independently),

discussing books read in the family, the presence and size of home libraries (both printed and electronic), the presence of devices with internet access and their frequency of use, preferred types of leisure activities, parents' education;

— Individual indicators of reader habitus, such as reading motives, reading preferences, perceptions of the image of a reader of printed books, electronic books, and content of social networks.

We conducted an empirical study of the reading practices of young people in the city of Yekaterinburg (Russia). The study was conducted from December 2021 to January 2022. In total, 200 respondents took part in the online survey. Table 1 shows that the age of the respondents is from 18 to 30 years. An online survey method was used for collecting information. The online survey made it possible to interrogate a sufficiently large set of respondents in a short time and make a general overview of the book-reading practices typical of young people. The selection of respondents was carried out using the "snowball" method, sending invitations to participate in the study through social networks ("VKontakte"), as well as by posting ads using the social network "VKontakte" in the chats of literary clubs in Yekaterinburg ("Lampa", "Vneklassnoe reading") and in the community "Atypical MARKETING" ("VKontakte").

**Table 1.** Gender and age of respondents.

| N | Gender | 18–22 Years Old | 23–25 Years Old | 26–30 Years Old | Total |
|---|--------|-----------------|-----------------|-----------------|-------|
| 1 | Men | 35 | 30 | 35 | 100 |
| 2 | Women | 35 | 30 | 35 | 100 |
| 3 | Total | 70 | 60 | 70 | 200 |

The obtained data were processed in the Vortex 10 program using such procedures as one-dimensional distribution (to characterize data on individual indicators) and two-dimensional distribution (to compare data on various indicators). When analyzing tabular questions, conditional indices were calculated according to the form of the arithmetic mean. The Cramer and Gamma coefficients were calculated to analyze the relationship in two-dimensional distributions, and the nonparametric Kruskal-Wallis's test and Student's *t*-test were also applied.

Limitations:

The study was conducted from December 2021 to January 2022 in Yekaterinburg (Russia). The structure of the study of reading habits, considering the approach of P. Bourdieu, includes the characteristics of readers' habits, the definition of the features of book reading practices, and their description. An online survey allowed characterizing the habitus of readers that are developing among the youth of Yekaterinburg. This study is based on quantitative methods. The advantage of quantitative research is the possibility of using statistical techniques to analyze data. This makes it possible to identify patterns and relationships between variables that might not be immediately apparent. However, there are also disadvantages to using a quantitative approach in sociology research. One of the main disadvantages is that it can be difficult to capture the complexity of social phenomena using numerical data. Social phenomena are often complex and multifaceted, and it can be challenging to capture this complexity using quantitative methods alone. Additionally, quantitative research can be limited by the types of questions that can be asked, and it may not be able to capture the subjective experiences of individuals.

## 3. Results

Habitus is a basis for the reproduction of practice, and it is laid from childhood. Therefore, it is necessary to find out how book reading was organized in the childhood of the respondents (Table 2). Considering the obtained data, the most popular childhood reading model is the combination of family reading and independent reading. This practice is characteristic of almost half of the respondents (49%). At the same time, almost a third

of respondents (29%) read books exclusively on their own in childhood. And only 10% of respondents noted that they did not read books in childhood. Thus, the main part of the respondents reproduced the practice of reading books from childhood, and at the same time, slightly less than 2/3 of them had the experience of group reading in childhood.

**Table 2.** Influence of parental education on childhood reading experience, % of respondents.

| The Education of the Respondent's Parents | | Childhood Reading Experience | | | |
|---|---|---|---|---|---|
| | | I Read Books with Parents | I Read with My Parents and on My Own | I Read Books on My Own | I Haven't Read Books |
| Mother's education | Secondary general education | 4.8 | 52.4 | 28.6 | 14.3 |
| | Secondary special education | 12.5 | 40.6 | 37.5 | 9.4 |
| | Higher education, incomplete higher education | 13.0 | 52.8 | 25.9 | 8.3 |
| Father's education | Secondary general education | 20.0 | 35.0 | 25.0 | 20.0 |
| | Secondary special education | 19.7 | 47.4 | 27.6 | 5.3 |
| | Higher education, incomplete higher education | 4.9 | 59.3 | 25.9 | 9.9 |
| | Total | 11.5 | 49.0 | 29.5 | 10.0 |

One of the objective factors that can affect reading in childhood is the education of parents. The reason is that it is believed that educated people have a high level of culture, and therefore they read more often and can teach their children to do so. The parents of the respondents generally have fairly high education. Approximately 90% of respondents have parents with an education above the secondary general level. Approximately half of the respondents have parents who have higher or incomplete higher education. Moreover, among mothers, the proportion of those with higher or incomplete higher education is higher than among fathers (56% of mothers, 49% of fathers). If we consider secondary specialized education, the opposite situation is observed since fathers (43%) proportion is higher than mothers (33%).

In general, the predominance of parents of respondents with higher education correlates with the fact that many respondents read books in childhood. However, a closer examination of the influence of parents' education on how they organized their child's reading in childhood reveals interesting results. As the mother's education increases, the share of those who always read books with their parents rises (from 5% to 13%), and the share of those who do not read books at all decreases (from 14% to 8%). This sampled result confirms the impact of parental education on the childhood reading experience. At the same time, the influence of the father's education paints a slightly different picture. With an increase in the father's education, the proportion of respondents who read books with their parents decreases significantly (5% compared to 20% among those whose father does not have a higher education), and the proportion of those who read both with their parents and on their increases significantly (from 35% to 59%). Thus, the high level of education of parents, in general, contributes to the introduction of the child to reading. However, at the same time, children whose fathers have higher education are more likely to join independent rather than family reading.

An important practice in introducing a child to reading is to discuss with him the books he has read. The data obtained show that 66% of respondents discussed books with their parents (16% of respondents always discussed, and 50% of respondents are from time to time). In general, this indicator correlates with the number of respondents who had family reading experience in childhood. However, it must be considered that they accounted for less than 2/3 of the total number of respondents. Therefore, it can be seen that at least some of the respondents who read books on their own had the experience

of discussing books with their parents. However, quarter of respondents (25%) did not discuss books with their parents.

The experience of reading books as a child is one of the foundations of the reader's habitus. It can be argued that approximately 2/3 of the respondents had good conditions for the subsequent reproduction of various reading practices, including family reading.

Another indicator that makes up the reader's habitus is the presence and volume of a home library, both printed and electronic. Almost all respondents (98%) have a traditional home library with printed books, while almost a third of the respondents do not have an electronic library. At the same time, Figure 1 shows that the respondents' answers were distributed approximately equally for the library in both printed and electronic formats. In both cases, the respondents mostly have few books in their home library; the most common volume is up to 50 printed books. This indicator was chosen by 35% for both printed and electronic libraries. Respondents rarely have larger home libraries. At the same time, there is an exception at the very edge of the scale since a fairly significant proportion of respondents have a home library with more than 200 print books. Thus, the respondents are dominated by small libraries, although many also have large libraries, which creates good conditions for reading.

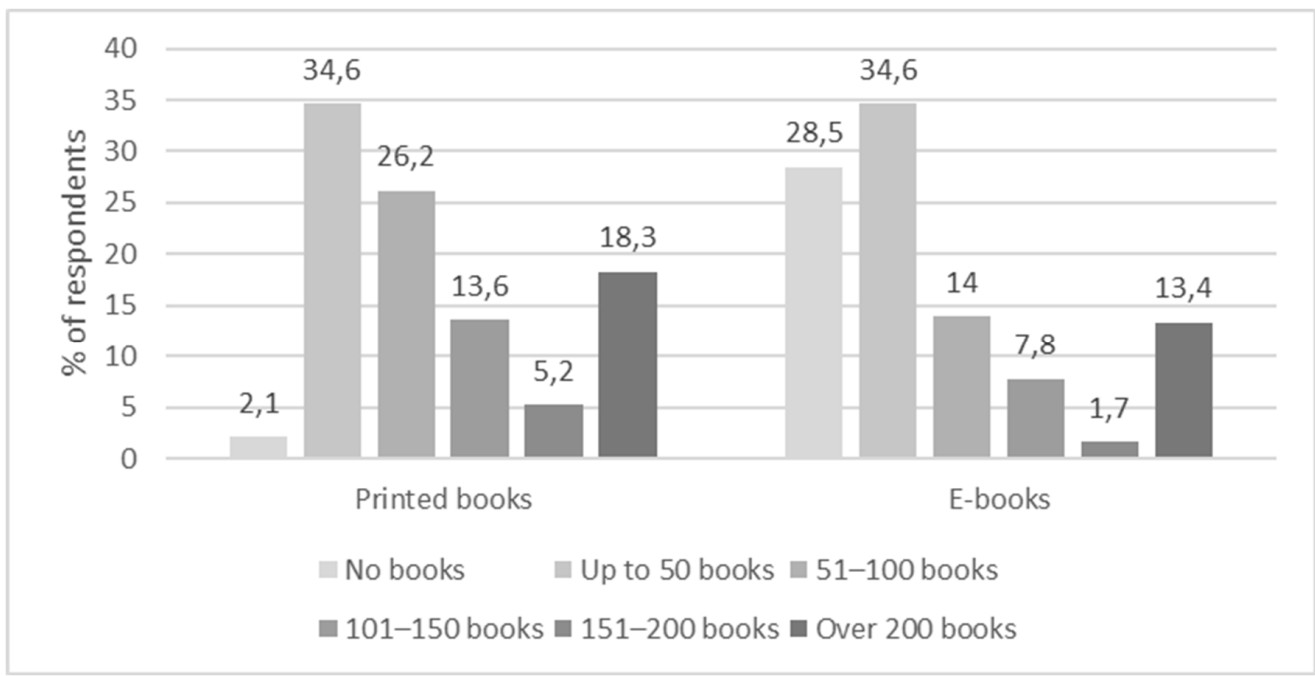

**Figure 1.** Respondents estimate the number of printed books in their home library, % of respondents.

It is equally important to find out what types of leisure time the respondents prefer and to single out among them the place of reading books. As can be seen from the data obtained (Figure 2), the top five most preferred leisure activities include watching videos or films (preferred by 80% of respondents), printed books (77%), social networks (70%), and hobbies (67%), as well as walking (55%). It is worth noting that books occupy a position even higher than social networks. In part, this shift may be because respondents knew the matter of the study, which could concentrate their engagement on reading books.

In addition to the proposed options, the respondents also offered their options for spending leisure time: "I am training a dog", "I work", "I study", "I am engaged in education", "I met friends and relatives", "Audiobooks from YouTube are something in between" videos" and "books". As you can see, the leisure time of the respondents is quite diverse. On average, the respondents chose about four ways to spend their leisure time, which, again, may indicate their activity and desire for diversity.

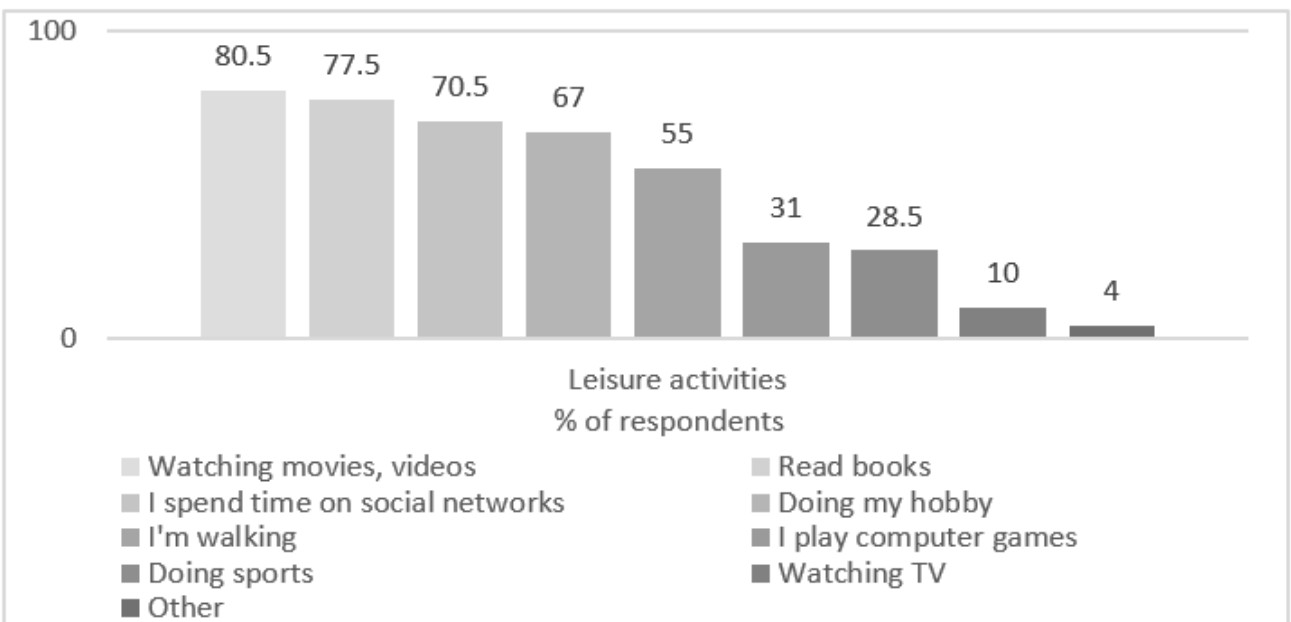

**Figure 2.** Preferred leisure activities according to respondents, % of respondents.

Another important characteristic is the frequency of using various electronic devices to access the Internet. The results showed that the majority of respondents (93%) use the Internet more than once a day, using a mobile phone for this purpose. More than half of the respondents (57%) also often use a computer or laptop. There are also significant groups of respondents who rarely use a computer or laptop (5% of respondents are once a week, and 9% of respondents are even less often). Finally, the tablet is the least used device for accessing the Internet; a total of 77% of respondents do not use it at all. Thus, the majority of respondents are active Internet users. However, the frequency of access to the Internet in itself does not indicate the purpose of its use. Therefore, it will be possible to consider in more detail the relationship between using the Internet and reading books when analyzing the frequency of reading various sources.

An important characteristic of the reading habit is the motivation of young people to read. Figure 3 shows that among the motives chosen by the respondents, the opportunity to plunge into the world of a book (for 77% of respondents), to find the necessary information (for 74% of respondents), or to relax, have fun (for 66% of respondents) is clearly in the lead.

Some respondents noted their motives in the "other" option: respondents read to broaden their horizons and enrich their speech, reading is interesting for them, it brings pleasure and new emotions, and makes it possible to experience the problems and feelings of another person. Among such answers, the following single indication of external motivation stands out noticeably: "I do not like to read, but I force myself because I feel the condemnation of society when I say that you do not read or do not like to read books".

In the last answer, the respondents indicated the influence of society on their motivation to read, which is consistent with the fact that a fairly significant proportion of respondents (42% of respondents) noted that they read books not of their own free will but out of necessity. Thus, even though mainly social motives (the ability to keep up a conversation about books or to feel superior to those who do not read books) appeared at the bottom of the list, we can take into account that respondents recognize the importance of external motivation.

An important characteristic of the reader's habitus is the reader's literary preferences. Based on the data obtained (Figure 4), we found that the top three preferred literary types include fiction (usually read by 92% of respondents), scientific or popular science literature (78% of respondents), and educational literature (54% of respondents). This fact may be because the main motives for reading for respondents are immersion in the world of books

and the desire for new knowledge. The most preferred types of literature correspond to these motives. The popularity of educational literature is also explained by the presence of a large proportion of students in the sample.

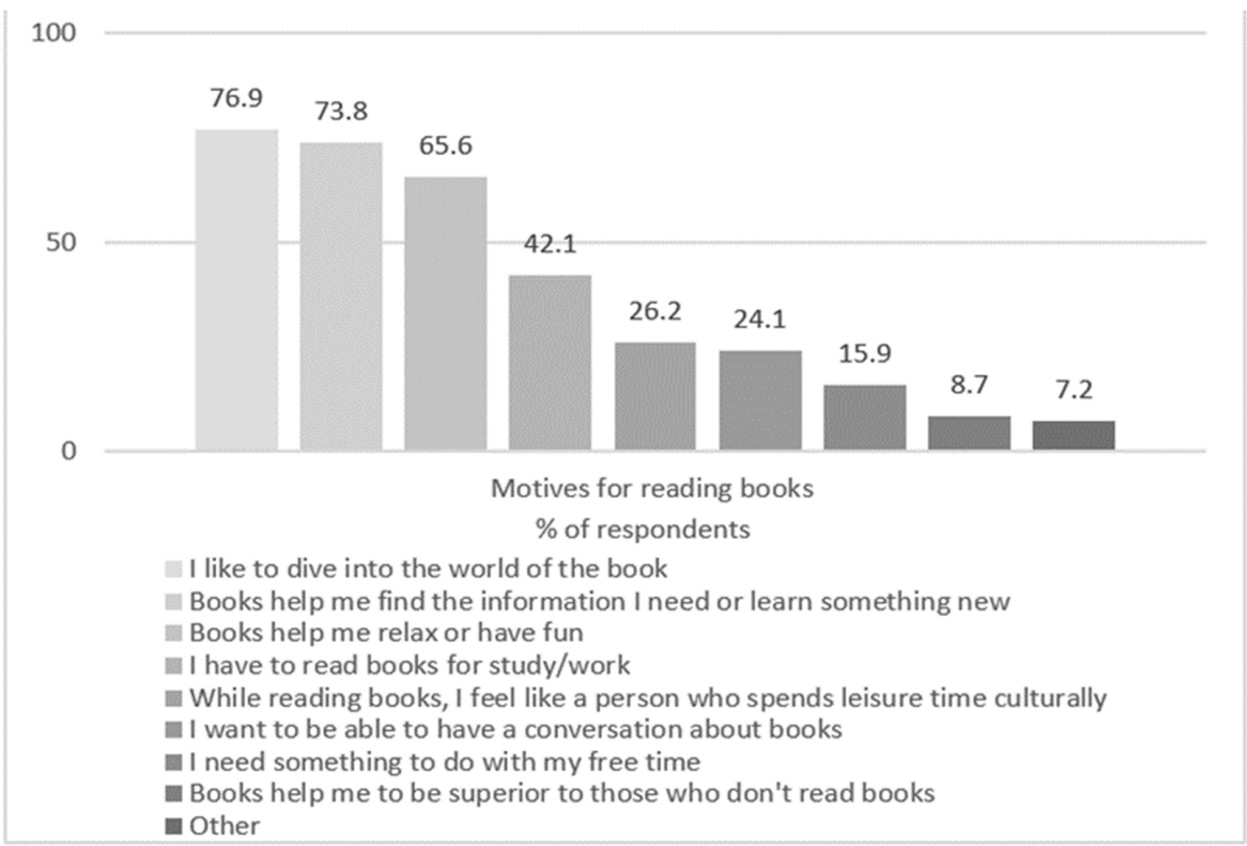

**Figure 3.** Motives for reading books among young people, % of respondents.

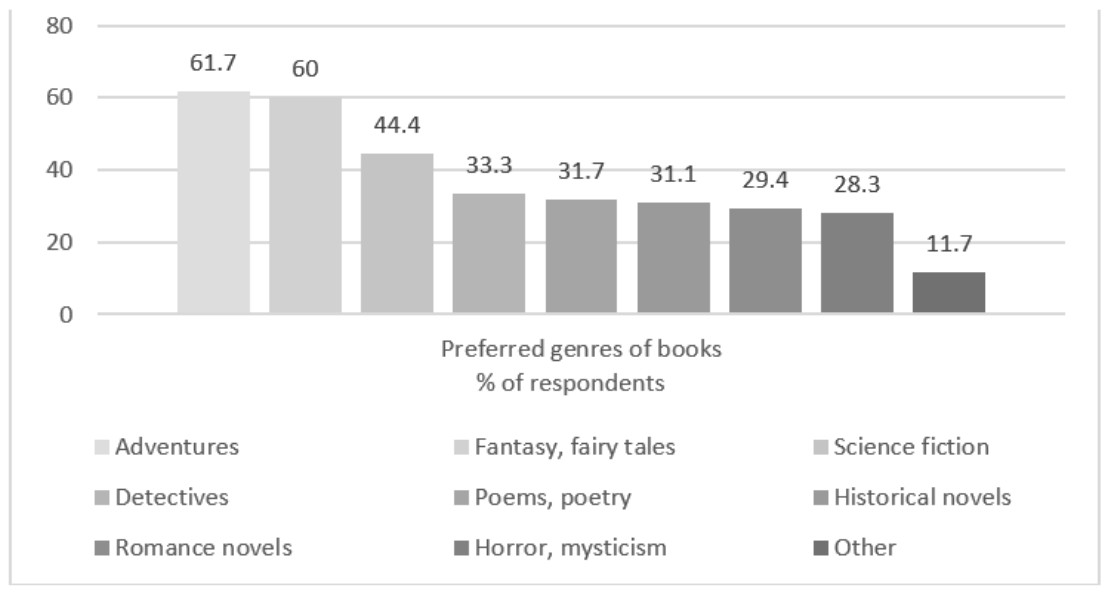

**Figure 4.** Preferred genres of books, % of respondents.

Slightly less than a third are the shares of respondents who read industrial and technical or reference literature (32% and 29% of respondents, respectively). The least popular are official documentaries, children's, and political literature. Also, some respondents

identified in addition to the above types of "biography" and "collections of interviews or essays". Considering the data on the preferred genres of fiction (Figure 4), one can notice that the genres of adventure and fantasy occupy the unambiguously leading positions (62% and 60%, respectively, chose these answers), and science fiction is slightly less popular (44% of respondents), and the rest genres are preferred by about a third of the respondents.

In the "other" option, the respondents identified classical literature, dystopia, short stories, modern prose, psychological/philosophical novel, literature about war, drama, comedy, literal role-playing games, and real role-playing games as separate genres. In addition, some respondents noted that they enjoy reading books with a mix of different genres and that they cannot accurately name the genre of their favorite books. In general, it can be seen that the range of reader preferences is quite diverse.

Another subjective component of the reader's habitus is the reader's perception of the image of various sources of information, which we examined using the example of reading printed and electronic books in comparison with reading the content of social networks (Table 3).

**Table 3.** Opinion of respondents about the image of the reader, % of respondents.

| N | Value Judgments of Respondents about Those Who Prefer a Certain Type of Information Sources | Sources of Information | | |
|---|---|---|---|---|
| | | Printed Books | E-Books | Social Networks (Posts and Blogs) |
| 1 | This is an interesting person | 50.5 | 35.5 | 12.0 |
| 2 | This person knows how to spend their free time properly | 32.5 | 23.5 | 7.5 |
| 3 | This is an erudite person | 30.5 | 22.0 | 5.5 |
| 4 | This is a modern, advanced person | 8.5 | 50.0 | 28.5 |
| 5 | This is a backward, ancient person | 3.0 | 0.0 | 4.5 |
| 6 | These are people who are wasting their time | 3.5 | 2.0 | 21.0 |
| 7 | It's a boring person | 3.5 | 2.5 | 3.0 |

The answers of the respondents were distributed curiously. Respondents endowed readers with negative characteristics very rarely. Only the reader of posts and blogs on social networks was rated by 21% of respondents as "a person who wastes time in vain". At the same time, the distribution of responses according to positive characteristics is more diverse. Respondents quite often marked the reader of printed books with positive characteristics. In total, 50% of respondents rated them as interesting people. Almost a third of the respondents noted that the reader of printed books is erudite and knows how to spend their free time. Only a few rated them as modern people, and in the section where the respondents express their suggestions, they identified them as people of the "old school". Respondents generally considered the e-book reader to be a modern, advanced, and also interesting person.

The reader of social network content deserved recognition only as a modern and advanced person, and the respondents rated their erudition and ability to manage free time quite low. It should be noted that when answering this question, quite a large proportion of respondents found it difficult to formulate their opinion about the reader. The share of those who found it difficult to answer is 19%, 24%, and 33% of respondents for printed books, e-books, and social networks, respectively. Other respondents argued in their answers that they cannot evaluate a person only by what sources of information people choose. For these respondents, preferred reading practices meant only that it was convenient for a person to read in a given format. The share of those who rated the reader as an ordinary person is 11%, 10%, and 5% for printed books, e-books, and social networks, respectively. Thus, for about a third of the respondents, the images of readers of various sources of information do not differ from each other.

Respondents were asked to give their assessments of the reader of printed books. Respondents called the reader of printed books an esthete who loves to touch and enjoy printed books, a rich person who "fills their world", and a lucky person "if a person can find everything in printed form". When assessing the e-book reader, the majority of respondents limited themselves to the proposed answers. Respondents added only such options: "Perhaps this person simply does not have the resources to be able to read printed books, whether financial or spatial" and "This is just a person who loves e-books or saves time, effort, and money on buying and carrying printed books, and for storage, by the way, too".

E-books are primarily regarded by these respondents as a more economical option, an alternative to printed books. Respondents described the reader of social network content as both neutral ("this person has little free time") and negative qualities, evaluating them as boring, poorly educated, and stupid people. Some respondents suggested that in social networks, such people can find useful information and not just "scroll the feed".

Thus the data highlights a positive image associated with readers of printed books and e-books among young people, with qualities such as being interesting, knowledgeable, and spending free time properly attributed to them. On the other hand, the perception of those who rely on social networks for information is more varied, with some positive attributes, such as being modern and advanced, as well as negative attributes, such as wasting time.

Figure 5 shows no difference in e-book reading frequency. This fact can be explained by the fact that in childhood, the respondents most likely read only printed books, and the reading experience in adulthood is affected only by reading in a familiar format. Thus, those who did not read printed books in childhood read them much less often in adulthood. A similar dependence on the example of audiobooks can be explained by the fact that reading with parents is a kind of "alive" audiobook format, and the habit of listening to the text of the work, and not reading it, can be more pronounced in those who in childhood used to listen to their parents reading a book rather than reading it yourself.

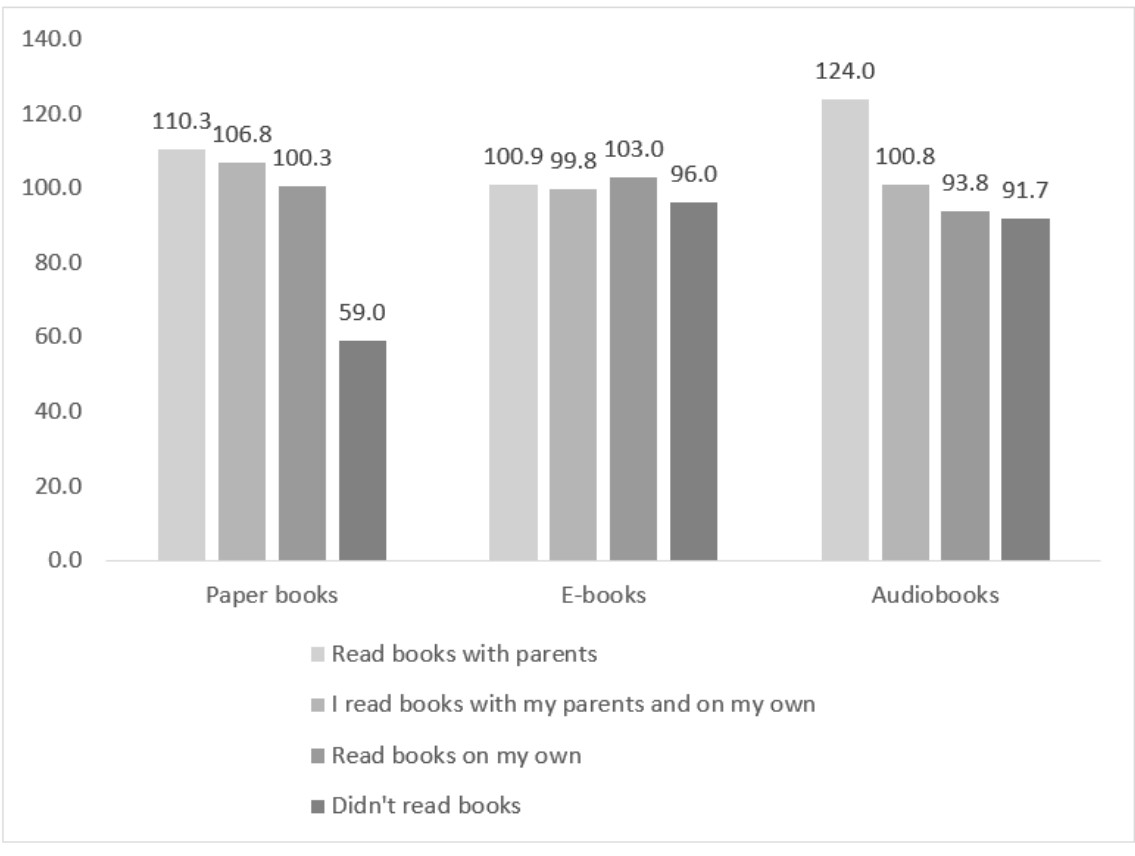

**Figure 5.** Influence of childhood book reading experience on the choice of book format.

A rather important aspect in studying the reading practices of modern youth is the preferred reading format. From the conclusions on the previous question, it is noticeable that printed books are read more often than books in other formats, and audiobooks are much less often, and the data obtained confirm the following conclusion: respondents mainly prefer printed and electronic format—91% and 75% respectively, and 44% of respondents listen to audiobooks.

In this study, we sought to understand why young people prefer one or another format of books. Therefore, we asked respondents to rate their preferred formats on a scale of 1–5 (where 1 is the lowest rating, and 5 is the highest rating) on eight criteria. Table 4 shows the average scores for each book format.

**Table 4.** Evaluation of various reading formats.

| N | Criteria of Evaluation | Printed Format | Electronic Format | Audiobooks |
|---|---|---|---|---|
| 1. | Comfort while reading | 4.2131 | 3.9669 | 3.8090 |
| 2. | Ability to read anywhere | 3.5082 | 4.5762 | 3.9438 |
| 3. | Habitually this format | 4.4044 | 3.4834 | 3.1124 |
| 4. | Availability of books in this format | 3.4317 | 4.3907 | 3.6292 |
| 5. | Convenience in the perception of the text | 4.5137 | 3.6026 | 3.3146 |
| 6. | The cheapness of books in this format | 2.1803 | 4.1854 | 3.8652 |
| 7. | Atmosphere, pleasant sensations | 4.5191 | 2.7881 | 3.6742 |
| 8. | No harm to eyes | 3.7923 | 2.6093 | 4.6854 |

The first criterion considered is comfort while reading. The data reveals that the printed format received the highest rating of 4.2131, followed closely by the electronic format with a rating of 3.9669. Audiobooks received a slightly lower rating of 3.8090. This suggests that young people generally find printed books to be the most comfortable reading format.

The second criterion is the ability to read anywhere. The electronic format received the highest rating of 4.5762, indicating that young people find it more convenient to read electronically. Printed books received a rating of 3.5082, while audiobooks scored 3.9438. This suggests that the mobility and portability offered by electronic devices make them a preferred choice for reading on the go.

Habitually reading in a particular format is the third criterion evaluated. The results show that young people tend to favor the printed format (rating of 4.4044) over the electronic format (rating of 3.4834) and audiobooks (rating of 3.1124). This indicates that the traditional habit of reading physical books has remained strong among young people.

The availability of books in a particular format is another important criterion. The electronic format scored the highest rating of 4.3907, indicating that young people find it easier to access books electronically. Printed books received a rating of 3.4317, while audiobooks scored 3.6292. This suggests that the wide availability of e-books and digital libraries has made electronic reading more appealing to young readers.

Convenience in the perception of the text is the fifth criterion considered. Printed books received the highest rating of 4.5137, indicating that young people find them more convenient for text comprehension. The electronic format scored 3.6026, while audiobooks scored 3.3146. This implies that the tactile experience of reading printed books contributes to a better understanding and enjoyment of the text for young readers.

According to the data obtained, it can be seen that printed and electronic formats are characterized by both high and low ratings. Audiobooks score neutral on most criteria. It is also noticeable that for each criterion, only one of the formats has a high score (more than 4). Moreover, for some criteria, a highly rated format can "cover" the shortcomings of another format. For example, respondents highly appreciate the atmosphere of printed

books, while for e-books, this is a clear disadvantage, but compared to the price of printed books, e-books are more affordable. Also, e-books of all three formats are the most harmful to the eyes, and the audiobook format avoids this problem. Thus, we can note that different formats of books are interchangeable, and the choice of one format or another depends to a greater extent on what exactly the reader considers the most important for himself. Also, perhaps the low popularity of audiobooks may be because they have few clearly expressed advantages over printed and electronic formats.

Differences in reading in different formats may be associated with a special type of reader's identity as follows [88]: a reader of printed books probably feels like a literate and intelligent person; an e-book reader is a person who finds time for intellectual leisure; an audio reader is a person who uses time efficiently, as well as an advanced user of digital gadgets (which can also be attributed to an e-book reader). According to the data obtained, the majority of respondents (48% of respondents) feel like people who find time for intellectual leisure. Other ideas about oneself as a competent and intelligent person, a reader who uses one's time efficiently, and an advanced user of digital gadgets are typical for more or less equal groups of respondents. In total, 18% did not identify themselves with any of the categories; they can be called undecided about the reader's identity.

## 4. Discussion

Traditionally, reading was mainly associated with printed books, requiring focused attention and imagination. However, with the proliferation of electronic devices, the landscape of reading has drastically changed. Many young people now turn to e-books, online articles, and social media platforms for reading material. These digital platforms offer interactive and multimedia features that enhance the reading experience, enticing the youth to engage more actively with texts. In the digital age, reading practices have undergone significant shifts due to the widespread adoption of digital texts. For example, in the book *Words Onscreen: The Fate of Reading in a Digital World*, Naomi S. Baron explores the implications of these changes and investigates how digital texts are influencing reading preferences, attention spans, and deep reading skills [89].

One of the most noticeable changes brought about by digital texts is the shift in reading preferences [89,90]. With the advent of e-readers, smartphones, and tablets, people now have a plethora of options when it comes to reading material. Traditional paper books are now being replaced by their digital counterparts, offering convenience and accessibility. The ability to carry an entire library in one device has led to an increase in the consumption of digital texts. However, N.S. Baron argues that this abundance of choices can also lead to a decrease in the commitment to reading a single text in its entirety, as readers may easily switch between different books or articles [89].

Another aspect influenced by digital texts is attention span. In the digital world, distractions are abundant. The constant availability of social media, notifications, and web browsing can divert readers' attention and hinder deep engagement with the material. N.S. Baron suggests that the nature of digital reading, characterized by skimming, scrolling, and multi-tasking, encourages a more superficial level of reading [89]. This shift in attention patterns poses a challenge to the development of deep reading skills, which require prolonged focus and concentration.

Despite these potential drawbacks, digital texts also offer new possibilities for engagement. Annotating, highlighting, and searching within a text is easier in the digital format, enabling readers to engage actively with the material. Digital texts also provide opportunities for interactive features such as hyperlinks, multimedia content, and collaborative annotations, enhancing the reading experience and potentially improving comprehension.

In the fast-paced digital world, where screens are ubiquitous, it is crucial to understand how youth engage with reading and how this engagement is shifting due to the advent of digital technologies. The transformation of reading practices has given birth to a new era, offering both opportunities and challenges for the youth in their pursuit of knowledge. Thus, in the book *Convergence Culture: Where Old and New Media Collide*, Henry Jenkins

explores the transformation of reading practices in the digital age and the implications it has for the younger generation's pursuit of knowledge [91]. This shift in reading habits has given birth to a new era that presents both opportunities and challenges for today's youth.

Traditionally, reading has been associated with printed books, but the advent of digital technology has expanded how young people consume information. Jenkins argues that in the era of convergence culture, where different media platforms intersect and interact with one another, the boundaries between reading and other forms of media are blurred [91]. Young readers now engage with content through a variety of mediums, such as e-books, online articles, blogs, and social media platforms.

One of the opportunities offered by this transformation is increased access to diverse sources of knowledge. With a few clicks, young individuals can explore a vast library of information on any given subject. They are no longer restricted to the knowledge found within the pages of a specific book but can access a spectrum of perspectives and opinions from various sources. This breadth of information allows them to deepen their understanding and make more informed decisions. However, Henry Jenkins shows that with this opportunity comes the challenge of evaluating the credibility and reliability of the information they encounter [91]. In the age of fake news and misinformation, young readers need to develop critical thinking skills to distinguish fact from fiction. The abundance of content available online requires them to be discerning readers who can analyze sources and fact-check information.

The digital reading experience has also impacted how young people interact with texts. Social media platforms and online communities provide spaces for readers to share their thoughts, interpretations, and critiques. This participatory culture allows for active engagement and collaboration, fostering a sense of community around shared interests. However, it also raises questions about privacy, ownership of ideas, and the influence of social validation on individual perspectives. Thus, Sonia Livingstone delves into the broader context of children's digital experiences and their impact on their reading habits [92]. Drawing on empirical research and theoretical frameworks from psychology and media studies, Livingstone highlights the intricate relationship between digital technologies, online platforms, privacy concerns, and their influence on children's engagement with reading materials. She critically examines how the digital world shapes youth's reading practices, the challenges it poses, and the potential benefits it offers.

The interaction of people with the text and the features that arise in this process in connection with the spread of the digital world is the subject of psychological research. For example, Maryanne Wolf explores the impact of digital reading on the way young people interact with texts [93]. With the rise of digital technology, the traditional reading experience has undergone significant changes, leading to both positive and negative effects.

One of the key impacts of the digital reading experience is the way it has transformed the way young people engage with texts [93]. Unlike traditional reading, which often involved the linear and deep reading of physical books, digital reading is characterized by skimming, scanning, and multitasking. The nature of digital media encourages a quick and superficial reading style, where users often jump from one piece of content to another, interrupting the depth and focus needed for in-depth comprehension.

The interactive features present in digital reading platforms have altered the way young people interact with texts. Digital books often include hyperlinks, multimedia elements, and social features, providing a more interactive experience. While these features can enhance engagement and motivation, they can also lead to distractions and reduced attention span. The constant availability of external stimuli and instant gratification in the digital reading experience can hinder the development of sustained attention and critical thinking skills.

The digital world has enabled youth to access a vast array of reading materials instantly. Social media platforms provide global access to a wide variety of genres, from literature to scientific research, catering to individual interests and preferences. This unlimited

accessibility has fueled curiosity and enabled self-directed learning, allowing young people to explore diverse topics that were previously inaccessible [94].

Furthermore, the digital world has facilitated new modes of reading through social media, where short-form content is prominent. Social media platforms encourage reading through brief and concise texts, challenging the youth to process information rapidly. This shift in reading practices may have profound implications for language comprehension and critical thinking skills.

However, the digitization of reading also poses challenges for youth. The constant presence of screens can be overwhelming and may contribute to shorter attention spans, reducing the ability to engage in deep reading and sustained analysis. Moreover, the ease of online access may lead to a desire for instant gratification and a lack of patience in reading lengthy or complex texts.

To address these challenges, it is essential to promote critical literacy skills in youth. Margaret Mackey's work *Literacies Across Media: Playing the Text* note that the concept of literacies beyond traditional printed texts as follows: literacy should not be limited to just reading and writing printed words but should encompass various forms of media such as digital texts, video games, and visual arts [95]. M. Mackey's work challenges traditional notions of literacy and encourages a more inclusive understanding of literacy practices in today's media-rich world. This article emphasizes the importance of considering the social and cultural contexts in which media literacy occurs. And this point of view correlates with the views of M. Mackey that individuals and communities negotiate meaning within specific cultural and societal frameworks.

The study of reading practices leads to questions and digital competencies of young people. Our study showed that a significant part of young people uses the Internet space for their purposes, both educational and entertaining. This issue is related to educational practice and the formation of digital literacy in the population. In this regard, this article is close to the article position of Sarva et al. (2023) focused on the development of digital competencies in students within the education field [96], Carroll et al. (2023) explored the concept of making digital technology education a community learning venture [97], Alneyadi et al. (2023) explored the effect of a digital learning environment compared to traditional methods on the literacy skills of fourth-grade students in the United Arab Emirates (UAE) [98], Al-Abdullatif and Alsubaie (2022) explored the utilization of digital learning platforms for teaching Arabic literacy in Saudi Arabia, specifically in the context of a post-pandemic mobile learning scenario [99].

A significant proportion of the information received by young people comes from the digital space. As the world becomes increasingly digitized, young people must develop strong digital competencies to thrive in this rapidly evolving landscape. Digital literacy, communication and collaboration skills, creativity, cybersecurity awareness, and digital citizenship are just a few of the crucial benefits that digital competencies offer. Young people proficient in digital competencies have a competitive edge in the job market. As industries continue to embrace digital transformation, possessing transferable digital skills will be critical for career growth.

Digital competencies also provide opportunities for entrepreneurship, freelancing, and remote work, allowing young people to access a wide variety of economic opportunities. This circumstance actualizes the issue of the formation and development of digital competencies of young people. Sarva et al.'s (2023) aimed to provide insights into the perspective of both students and stakeholders regarding the importance of digital competencies and how they can be fostered [96]. The findings of the study reveal that both students and stakeholders acknowledge the significance of digital competencies in the education field. These competencies are seen as essential for enhancing student learning, improving teaching practices, and preparing students for their future careers. However, the study also highlights the existing gaps in students' digital competencies and the need for interventions to bridge these gaps. Thus, Sarva et al.'s (2023) shed light on the importance of digital competencies in the education field from the perspectives of students and

stakeholders [96]. It underlines the need for interventions to develop these competencies and provides recommendations to enhance their incorporation into educational practices.

The present study presents the factors that determine the choice of reading formats by young people. At the same time, it is shown that the experience of reading books in childhood and the influence of parents play a significant role in reading at an older age. In this aspect, this study correlates with the study of early childhood literacy development by López-Escribano et al. (2021), who explored the effects of e-book reading on the emergent literacy skills of young children [100]. Emergent literacy refers to the development of skills that lay the foundation for reading and writing. López-Escribano et al. (2021) aimed to analyze previous studies to determine the impact of e-book reading on emergent literacy skills in young children [100]. They specifically focused on the cognitive, linguistic, and socio-emotional aspects of literacy development. The study considered children between the ages of 2 and 8 years old, investigating the effects of e-book reading on alphabet knowledge, phonological awareness, print knowledge, vocabulary development, and comprehension skills.

The research conducted by López-Escribano et al. (2021) revealed that e-book reading has a positive impact on young children's emergent literacy skills. E-books offer interactive and multimodal features that enhance alphabet knowledge, phonological awareness, print knowledge, vocabulary development, and comprehension skills. These findings highlight the potential of e-books as beneficial tools for promoting early literacy development in young children.

Educators and parents need to facilitate a balance between digital reading and engaging with printed books, as both offer unique benefits [95,100,101]. Encouraging discussions and providing opportunities for reflective reading can help develop deeper thinking skills, regardless of the medium. Thus, M. Mackey explores the intersection between youth practices of reading and the digital world and examines how young people engage with different forms of media, including computer games, print books, and digital storytelling, and the implications for literacy development [95]. Through extensive research and analysis, M. Mackey provides insights into how digital technologies shape young individuals' reading practices, comprehension skills, and overall literacy experiences. Her interdisciplinary approach combines education, media studies, and literacy research to provide a comprehensive understanding of the complex relationship between youth, reading practices, and the digital landscape.

The consequences of the research on the youth reading experiences highlight various aspects to consider when promoting a love for reading among people. Parental education emerges as an influential factor in shaping reading habits, emphasizing the importance of parental involvement and support. The preference for both printed and electronic libraries underscores the need for diverse reading platforms to accommodate individual preferences. The popularity of reading printed books, even in the digital era, demonstrates the enduring value of this traditional format.

Understanding the sociological factors that shape reading experiences is essential in fostering a love for reading in the youth and promoting literacy in society. Further research can delve deeper into the nuances of parental influence on reading habits, explore possible gender biases, and investigate the impact of educational interventions on reading preferences and habits.

## 5. Conclusions

Sociological research plays a crucial role in understanding the factors that shape individuals' reading experiences. The findings of this sociological research shed light on the influence of parental education on childhood reading experiences. The study reveals that parents with higher education levels tend to foster a reading culture within their families. Printed books remain highly valued as a leisure activity, even in the era of digital media dominance. Motives for reading encompass a range of psychological, intellectual, and leisure needs, highlighting the multifaceted role reading plays in individuals' lives.

Lastly, the preference for fiction, scientific literature, and educational texts showcases respondents' inclination towards both entertainment and knowledge acquisition.

Reading is an important way for young people to acquire culture. Reading allows young people to explore different perspectives and ideas, develop critical thinking skills, empathy, and understanding for others, and engage with the world around them. Reading practices are a dynamic system that depends on different contexts and may include reading from a wide variety of sources. However, for an adequate understanding of the text, people need to be educated, so the issues of the continuity of the habit of reading from parents to children, and the acceptance of reading as a form of one's activity are important. Reading is associated with the technique of perception, presentation, and comprehension: "By being educated in a technique, we are also educated to have a way of looking (Betrachtungsweise) which is just as firmly rooted as that technique" [102]. People do not just see the text; they need to learn to see it in its semantic, cultural, and historical aspects.

The practice of reading depends both on individual characteristics of the individual and external conditions. Therefore, it is necessary to consider both the characteristics of the reader and the characteristics of the social reality in which the studied practices unfold. The reader's habitus model revealed the following characteristics:

— Objective characteristics of the reader's habitus: reading books in childhood, subsequent discussion of the books read in the family, the presence and volume of a home library, the availability of devices with the ability to access the Internet, and the frequency of their use, preferred types of leisure, parental education;
— Subjective characteristics of the reader's habitus: assessment of the reader's image of different sources of information, reading motives, and reader preferences.
— The study of the objective components of the habitus of the reader on the example of the youth of Yekaterinburg shows the following:
— In childhood, most respondents had the conditions for becoming an active reader in the future (almost all of them (90%) were introduced to reading, and about 2/3 of the respondents were involved in family reading practices);
— In the current situation, many people have constant access to the Internet, but not everyone has an electronic library (28% of respondents do not have an electronic library);
— In general, the respondents are dominated by small home libraries (about half of the respondents have a home library of fewer than 100 books, both print and e-books).

At the same time, the subjective components of the reader's habitus indicate that the respondents have a wide range of motives for reading, with a predominance of internal motivation, the desire for reading on the part of the individual, and not under compulsion. Also, readers have a variety of reading preferences, and they positively evaluate the image of the reader of the book in the traditional format.

The respondents avoid sharply negative judgments and strive for neutrality in assessing the image of the reader of various sources of information. They recognize the insignificance of people's preferences in reading when assessing their personality. However, readers of e-books and print books were more positively described by respondents, which may reflect perceptions of a reader of books in any format as a more interesting, educated, and worthy person compared to someone who reads exclusively social media content. The characteristics of the reader of the present time include the volume of the home library and the frequency of access to the Internet. Regarding the present moment, the conditions that the respondents themselves create for themselves are already less conducive to reading books.

One of the advantages of print books is that they offer a tactile experience that digital books cannot replicate. The feel of a book in your hands, the sound of turning pages, and the smell of ink and printing all contribute to a unique reading experience that cannot be replicated by digital books. Additionally, print books do not require any special technology to access and can be read anywhere, anytime, without the need for an Internet connection. On the other hand, digital books offer a level of convenience that print books cannot match. With digital books, readers can carry an entire library with them on their devices, making it

easy to access books on the go. Additionally, digital books are often cheaper than printed books and can be purchased and downloaded instantly. However, digital books have some disadvantages as well. They require a device to access, which can be a barrier for some readers, especially those who do not have access to the Internet or cannot afford a device. Additionally, reading from a screen can cause eye strain and fatigue, making it less enjoyable for some readers.

The subjective components of habitus make it possible to notice that the respondents have a wide range of motives for reading. Readers have a pronounced intrinsic motivation. They tend to read themselves and have a variety of reading preferences. These characteristics are also combined with a fairly common point of view that ideas about the image of the reader do not depend on what source of information this person reads. Although at the same time, the image of a book reader is rated as more positive than the image of a social network reader.

Age and gender differences have little effect on the reading habits of different groups of young people, but there are also certain relationships. For example, not only are men less likely to read books as children, but they are also more likely than women to not discuss books with anyone as adults. Thus, the habitus of reading can be understood as a form of life since it includes the following components: rules of behavior, for example, a certain style of expressing emotions and actions, and a component of knowledge, cultural and historical context that defines the way of life and lifestyle: value structures, type of production, beliefs, and ways of organizing personal space and time.

**Author Contributions:** Conceptualization, A.S. and A.N.; methodology, A.N. and A.S; validation, investigation, A.S., E.K. and A.N.; resources, A.S. and A.N.; data curation, E.K.; writing—original draft preparation, A.S., E.K. and A.N.; writing—review and editing, A.S., A.N.; visualization, E.K.; project administration, A.S. All authors have read and agreed to the published version of the manuscript.

**Funding:** This research received no external funding.

**Institutional Review Board Statement:** The study was conducted in accordance with the Declaration of Helsinki and approved by the local scientific PhD committee (protocol number: 0041598).

**Informed Consent Statement:** Informed consent was obtained from all subjects involved in the study.

**Data Availability Statement:** Not applicable.

**Conflicts of Interest:** The authors declare no conflict of interest.

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
