# Peer review of "Youth Practices of Reading as a Form of Life and the Digital World"

_societies, doi:10.3390/soc13070165_

Round 1

Reviewer 1 Report

Overall I find the theoretical claims the paper makes to be way overstated compared to the empirical analyses. Moreover, it is highly unclear what the goal of the empirical analysis actually is and how this connects to the theoretical focus od the paper.

First, the paper makes several very grand theoretical claims throughout. E.g. in the abstract "The development of digital technologies leads to the transformation of the socio-cultural life of a person" and "The emergence of new forms of reading translates the text from the usual paper into an electronic format, which transforms the way of life". It is neither clear to me what the basis for such grand claims are or how they are relevant to the study. Further, a couple of examples from the introduction: "reading is a way for people to acquire culture" - no citations support this claim and what does it even really mean?. "one of the significant pedagogical issues is the question that refers to philosophy in its essence: how to teach to read social and cultural meanings in the text?" - again, how is this at all related to the empirical study?

My main and overall concern is that it is not clear what empirical question the paper is trying to answer, and how this relates to the many theoretical directions the paper lists in the introduction.

- The implication of this is also that the empirical analysis reads as a list of different survey questions. It is not clear how the analysis of each question connects to each other and to the papers main purpose (which is not clear either).

- E.g. the paper mooves from retrospective childhood data to adult data without giving much notice to it.

- Given that the title of the paper is about the digital world I was suprpriced of the little attention given to of E-books transforms the practice of reading. The statement in the introduction "This article aims to study young people's reading practices in the modern digital world" seem to give the impression that digital reading practices would be in focus. To me, only the questions about home libraries and Table 4 relate to this.

- For the question about home libraries a discussion of whether this question even makes sense for digital collections is needed. Personally, I don't have a "home library" of E-books - I "rent" them as a subscription service and only have one or two downloaded at a time.

- Given the stated purpose of analyzing digital reading practices I was missing information on whether you read more/less with E-books, do you read different types of books?, read in a different way? for a different purpose? etc.

Author Response

Dear Reviewer,
Thank you for your appreciation of our work and your comments. Your comments helped us a lot in finalizing the article and clarifying our position.
We have specified the purpose of the article. Thank you for this comment. The reformulation avoided ambiguity.
We have expanded the "Introduction" section. We have added conceptual links that are relevant to the issues at hand.
The main purpose of the article is to show through which channels young people receive information that meets the goals of young people. We also start from the idea that previous experience influences present choices. Therefore, questions about childhood are also raised in this study.
Question about home libraries. It is typical for the Russian space that books were collected and passed down from generation to generation. This practice does not negate the fact that we mostly read electronic magazines and books today. Therefore, the discussion of this issue reflects the mentality of people, that is, this issue corresponds to the logic of the study.
We have adjusted the goal, It now corresponds to the research. Therefore, the issue of purely digital reading is not considered by us. However, there is a comparison of reading format selection criteria (Table 4).
Once again, I want to express my gratitude to you, because your comments contributed to a clearer presentation of the idea of ​​the article.

Reviewer 2 Report

This is a valuable topic to look into and the findings provide some quantitative insight into the lives of young readers. I would advise extending the literature review section to include more reference to recent studies which investigate the reading lives of youths. Also justify why you have chosen a quantitative approach over qualitative? How were participants chosen?  Make these questions clearer. A fantastic study, well done.

Author Response

Dear Reviewer,
Thank you for your appreciation of our work and your comments. Your comments helped us a lot in finalizing the article and clarifying our position.

We have expanded the "Introduction" section. We have added conceptual links that are relevant to the issues at hand.
We used an online survey. Therefore, we did not have spoiled or unfinished questionnaires. The survey was anonymous. We connected our students, and they already know each other. This chain of relationships is not rendered.
We agree that research can be conducted using either a quantitative or qualitative approach. Both methods have their advantages and disadvantages, and the choice of approach depends on the research question, the research design, and the data analysis techniques.
In the "Limitations" section, we have outlined our position and the limitations of our method. During the research, we paid attention to a qualitative interview. But this study is a separate study that does not fit into the presented one in terms of scope and goals.

Reviewer 3 Report

The connection between the theoretical framework and the data from the empirical study should be made stronger.

Please check the sentences in lines 75-77 and 79-80 and 186.

Sentence repeated in 235/236

328: hobbies and hobbies 

Figure 2 is hard to read

line 338/339: quotation marks 

520 Internet - internet

Author Response

(The authors gave the same response as above.)
